# Automated Machine Learning algorithm for Kidney, Kidney tumor, Kidney Cyst segmentation in Computed Tomography Scans

Vivek Pawar[1], Bharadwaj Kss[1]

Endimension Technology Private Limited (Incubator Under SINE IIT Mumbai)1

vivek_pawar@endimension.com

**Abstract.** In this paper, we have described an automated algorithm for accurate segmentation of kidney, kidney tumors, and kidney cysts from CT scans. The Dataset for this problem was made available online as part of KiTS21 Challenge. Our model uses a 2 stage cascaded Residual Unet architecture. The first network is designed to predict (Kidney + Tumor + Cyst) regions. The second network predicts segmented tumor and cyst regions from the output of the first network. The paper contains implementation details along with results on the test set.

**Keywords:** KiTS21, Kidney Segmentation, Unet

## 1   Introduction

There are more than  400,000  new cases of kidney cancer each year[1],  and surgery is its most common treatment [2]. KiTS21 challenge [3] was conducted to accelerate the development of reliable kidney and kidney tumor semantic segmentation methodologies. Ground truth semantic segmentation for abdominal CT scans of 300 unique kidney cancer patients were provided as part of the training dataset for the challenge. The submission models are then evaluated on a test set of 45 patients (part of 300 CT scans) which is separate from the official test set of 100 cases.

## 2   Methods

### 2.1   Training and Validation Data

Our submission made use of the official KiTS21 training set alone. We divided the official KiTS21 dataset into training, validation, and test set. We used a validation set to finetune our approach. The approach which had the best score on the test set was used to create the final submission. The test and validation set are sampled from 300 cases, initially provided in the KiTS21 challenge.
The below table shows the distribution of scans among different sets.

| Training | Validation | Test |
|----------|------------|------|
| 225 | 30 | 45 |

Table 1: Distribution of available samples between training, validation and test set

## 2.2 Preprocessing

We use a three-stage segmentation process, the first stage for Kidney+Tumor+Cyst segmentation. For the first stage, we resample all cases to a common voxel spacing of 3.22×1.62×1.62 mm, with a patch size of 80X160X160, for the training cases. The voxel spacing 3.22×1.62×1.62 mm was chosen on the basis of the median spacing of the training dataset which is 3.22×0.81×0.81mm. The x and y spacing was doubled to avoid training large numbers of patches. Before creating the patches for training we cropped the abdomen region from the entire CT scan to avoid training of unnecessary negative patches. We used bicubic interpolation to resample the cases.

Each case is then clipped to the range [−80,304]. We then subtract 101 and divide by 76 to bring the intensity values in a range that is more easily processed by CNNs. The clipping range was selected after analyzing the voxels covered by kidney, tumor, and cyst segmentation in the CT scans over the entire training set. The -80 and 304 values are simply the values at 3 percentile and 97 percentile of the distribution respectively. Similarly, 101 and 76 are the mean and standard deviation of the distribution.

For the Second Stage, we resampled the voxel spacing to 0.78X0.78X0.78 mm and used the patch size of 64X128X128. Since we had out of Kidney+Tumor+Cyst Segmentation output from the first stage we only used the patches where the first network predicted any output. Similar to the first stage, We clipped voxels to the range between [-31, 208], then subtract by 55 and divide by 65 based on the voxel covered by Tumor and Cyst segmentation in CT scans.

Similarly, the third stage segments the cyst part of the scan. Voxel spacing and clipping range are the same as stage 2. We changed the input patch size to 32X64X64 to capture sufficient contextual information for the cyst.

## 2.3 Network Architecture

For both stages, we used base 3D unet architecture (as illustrated in Figure 1) with residual blocks (Figure 2). Both Networks use 3D convolutions, ReLu nonlinearities, and Instance Normalization instead of batch normalization. We double the number of feature maps with each DownBlock in the Unet. For the Upsampling block, we used the Scale of 2 to upsample input.

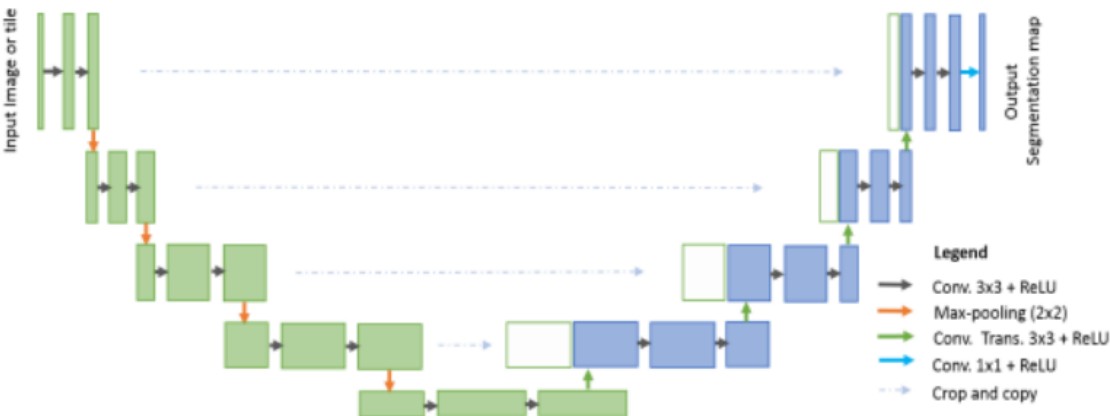

Fig. 1. Base U-Net architecture with convolutional encoding and decoding units that take the image as input and produce the segmentation feature maps

This architecture uses residual blocks instead of simple convolution sequences, which is implemented in a similar fashion as conv-instancenorm-relu-conv-instancenorm-relu. The addition of the residual takes place before the last relu. We base our implementation on nnU-Net [4].

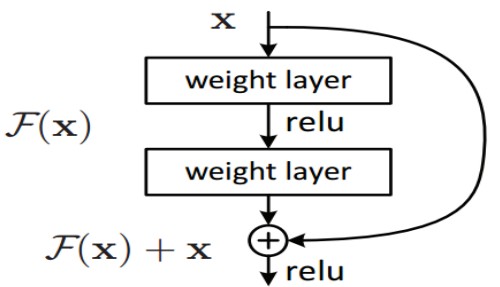

Fig. 2. Residual Convolution Block

## 2.3 Network Training

All the stages are trained with Adam optimizer and intial learning rate of 1e-3. We reduce the learning rate if the loss did not improve within the last 10 epochs, we reduce the learning rate by a factor of 5. We stopped the training if the loss did not reduce in the last 20 epochs. The batch sizes for stage 1, stage 2, and stage 3 are 2 and 4, 32 respectively. We used flipping, rotation, scaling, brightness, contrast, and gamma augmentations to augment patches during training. Loss function was a combination of cross-entropy and dice loss. The training was done on Nvidia GeForce GTX 1080 Ti GPUs. All networks were implemented with the PyTorch framework.

## 3  Results

We did not experiment with different network architectures. We tested our networks on test data consisting of 45 samples. The evaluation metric uses the same setup as challenge evaluation. It consists of two metrics Sørensen-Dice and Surface Dice [5]. The score for the kidney is computed by treating all the labels i.e. kidney, tumor, and cyst labels as foreground and rest as background. Similarly Mass consists of labels from tumor and cyst. Our network is very effective at detecting both kidney and tumor in most cases. However, the segmentation of cysts needs improvement. We noticed that the cyst network fails to segment smaller size cyst areas in CT scans especially if the size is smaller than 10X10 which ranges to 4 slices.

|  | Kidney | Mass | Tumor |
|---|---|---|---|
| Dice | 0.96 | 0.82 | 0.77 |
| Surface Dice | 0.93 | 0.71 | 0.67 |

Table 1: Experiment results

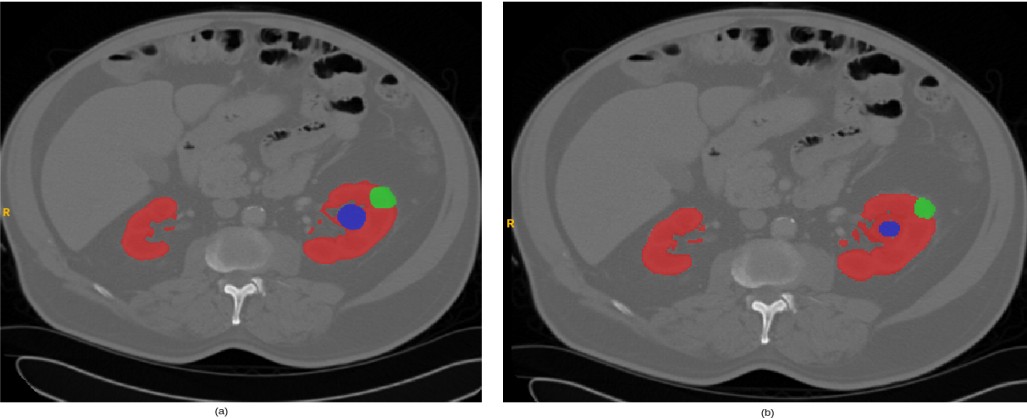

Fig. 3. Experimental outputs from our network. The first image shows the ground truth on a slice. The second image shows network predictions.

## 3  Discussion and Conclusion

In the preprocessing section of the paper, we mentioned that we used the same clipping values as the second stage which has tumor and cyst voxel values. Surprisingly using values based on voxels covering cyst does not yield better results for cyst segmentation.

In conclusion, We described a three-stage semantic segmentation pipeline for kidney and tumor segmentation from 3D CT images. Internal evaluation results on KiTS21 challenge results are 0.96, 0.82, and 0.77 average dice for kidney, mass, and tumor respectively.  We will also include the performance on the official test set once available.

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
