# OpenReview forum: "Automated Machine Learning algorithm for Kidney, Kidney tumor, Kidney Cyst segmentation in Computed Tomography Scans"
_MICCAI.org/2021/Challenge/KiTS — Submitted to KiTS21 Challenge_

### Official Review · Reviewer_rQy3 · 2021-08-30

**Rating:** 5

**Review:**

The authors provide a fairly brief paper describing a two-stage approach based on 3D Residual U-Net. Many important details are missing, and the authors should consult the provided template for a list of things that should be addressed within the paper for their revision.

---

### Official Review · Reviewer_J57R · 2021-08-30

**Rating:** 4

**Review:**

### Overall

- Architec-ture is hyphenated unnecessarily
- Please use the capitalization "KiTS21" where applicable

### Introduction

- It might be a good idea to clarify that the 45 cases are just a validation set, and that there is an official test set of 100 cases

### Methods

- Which algorithm did you use to do the resampling?
- How did you choose your range to which to clip the HU values?

### Results

- Did you experiment with any other models? What were their performances? Did you benchmark your algorithms based on surface dice as well? Did you notice any common error modes?
- Please make sure to include the official results as well once they are known

### Discussion and Conclusion

- Please add a discussion/conclusion section to summarize your approach and results

---

### Decision · Program_Chairs · 2021-08-30

**Decision:**

Major Revisions

**Comment:**

Please address the reviewer comments and resubmit